# Enhanced Low-Velocity Impact Resistance of Helicoidal Composites by Fused Filament Fabrication (FFF)

**DOI:** 10.3390/polym14071440

**Published:** 2022-04-01

**Authors:** Xiaochun Lu, Xiameng Zhang, Yangbo Li, Yan Shen, Yinqiu Ma, Yongdong Meng

**Affiliations:** College of Hydraulic and Environmental Engineering, China Three Gorges University, Yichang 443002, China; luxiaochun1014@163.com (X.L.); zhangxiameng78@163.com (X.Z.); sh18727276638@163.com (Y.S.); m1017870220@163.com (Y.M.); meng@ctgu.edu.cn (Y.M.)

**Keywords:** biopolymer, PLA-Based laminates, impact behavior, 3D printing, fused filament fabrication (FFF)

## Abstract

Bioinspired composites, capable of tailoring mechanical properties by the strategy of making full use of their advantages and bypassing their drawbacks, are vital for numerous engineering applications such as lightweight ultrahigh-strength, enhanced toughness, improved low-/high- velocity impact resistance, wave filtering, and energy harvesting. Helicoidal composites are examples of them. However, how to optimize the geometric structure to maximize the low-velocity impact resistance of helicoidal composites has been ignored, which is vital to the lightweight and high strength for aerospace, defense, ship, bridge, dam, vessel, and textile industries. Here, we combined experiments and numerical simulations to report the dynamic response of helicoidal composites subjected under low-velocity impact (0–10 m/s). Our helicoidal structures, inspired by the Stomatopod Dactyl club, are fabricated using polylactic acid (PLA) by FFF in a single-phase way. The helicoidal strategy aims to exploit, to a maximum extent, the axial tensile strength of filaments and simultaneously make up the shortage of inter-filament contact strength. We demonstrate experimentally that the low-velocity impact resistance has been enhanced efficiently as the helicoidal angle varies, and that the 15° helicoidal plate is better than others, which has also been confirmed by the numerical simulations. The findings reported here provide a new routine to design composites systems with enhanced impact resistance, offering a method to improve impact performance and expand the application of 3D printing.

## 1. Introduction

Impacts occur in such cases as space debris impacting spacecraft surface, birds, or hail hitting airlines, shock waves impacting soldiers’ helmets, vehicles colliding with submarine tunnels, and missiles attacking dams/bridges. If the protective shield materials do not have adequate dynamic impact resistance ability, catastrophic consequences will take place, such as spacecraft explosion, airline crashes without passengers’ survival, veterans’ psychological and physical suffering, submarine tunnel leakage, and dam failure flood. The Space Shuttle Columbia Disaster is a typical example [1]. Pieces of isolating foam, which separated from the left bipod ramp area on the external fuel tank, collided with the Columbia at a relative velocity of about 877 km per hour. The collision damaged the panel of the carbon heat shield material on the orbiter’s left wing. Therefore, it is significantly meaningful to study enhanced dynamic impact resistance by all kinds of ways.

In order to improve the dynamic impact resistance of shield materials, researchers and engineers have started to turn their eyes on cutting edge materials, such as bio-inspired composites [2] or mechanical metamaterials [3,4,5]. Their strategy of dynamic mechanical enhancement is to make full use of their advantages and bypass their drawbacks, by means of learning from nature [6]. As more and more natural evolution wisdoms have been exposed, bio-inspired composites have sprung up, such as nacre [7,8], honeycomb [9], bone osteon, conch shell, and the Stomatopod Dactyl club. They have played a vital role on mechanical properties’ enhancement, for instance, strength [10,11], toughness [12,13,14], low or high velocity dynamic performance [15,16,17], energy absorption [12,18], and acoustic/elastic wave transmission control [19,20]. Helicoidal composites [2,21,22,23], inspired by the Stomatopod Dactyl club, are examples of them. In helicoidal composites, the layers of locally parallel filaments have laminated each other by the style that each adjacent layer is skewed by a constant angle from the layer below it, as shown in Figure 1a,b. Using the contact among filaments to simulate the bonding of the soft matrix of the lobster cuticle, the axial strength of filaments is much higher than the contact strength between filaments. After undergoing dynamic impact loading, regular filament laminates usually appear to be anisotropically destroyed. To avoid damaging where the weakness is, the helicoidal composites are naturally evolved as described above. By skewing a certain angle between filaments laminates, the helicoidal composites can be presented as quasi-isotropic as long as they are featured by an appropriate skewing angle, which is also a key content of this research. As such, helicoidal composites as filament-wound pressure vessels [24], carbon fiber-reinforced composites, etc. have been applied to advance strength, fatigue and corrosion resistance in aerospace, military applications and hydrospace.

Current investigations into the static mechanical improvement of helicoidal composites range from shear wave filtering [25], stiffness and toughness [26], to twisting cracks [27,28]. Recent literatures [29,30,31,32,33,34,35,36] on the low-velocity impact have been conducted upon commercial carbon fiber laminates. However, the optimal helicoidal angle resisting the low-velocity impact testing has not been worked out yet [37]. How to optimize the geometrical structure to maximumly utilize the components’ strength limit or maximally enhance the impact resistance has always been tremendously demanded in aerospace engineering, because it is the only way to solve the conflict between lightweight and high strength, especially in aerospace.

Fused Filament Fabrication (FFF) has popular application in education, exhibition, and research because of the capability of fabrication in complicated and flexible structures [38]. However, the anisotropy and mechanical shortage handicap extensive engineering applications, especially in structures enduring the key loads. Moreover, higher printing temperatures, suitable printing speed, and thinner printing thickness are important factors for the quality performance of the 3D printed models in this paper [39,40]. The bionic helicoidal structures might provide the potential solution to enhance the FFF product load-carrying capacity. To validate that the helicoidal strategy is effective in enhancing dynamic impact resistance, we employed the FFF to fabricate our experimental specimens. In addition, in order to reduce the influence of the printing method of FFF on the test results, we selected the appropriate optimized 3D printing parameters.

In order to optimize the geometrical structure that maximumly utilizes the components’ strength limit or maximally enhances the impact resistance, we have carried out low-velocity impact tests and numerical simulations using the nonlinear finite element code, LS-Dyna. We have compared the dynamic response of circular plates with different helicoidal angles to investigate the underlying mechanism of the enhanced impact resistance. The numerical simulations are aimed to prove the validity of experimental tests. In order to conduct numerical simulations, the static axial tensile test has also been implemented to obtain the stress–strain curves and Young’s Mmodulus.

## 2. Materials and Methods

### 2.1. Microstructure Design and Specimen Fabrication

In order to explore how the helicoidal angle affects the dynamic performance, we designed a series of round plate specimens with a diameter of 150mm with helix angles including 0°, 15°, 30°, 45°, 60°, 75° and 90°, which are fabricated using polylactic acid (PLA) by a Fused Filament Fabrication (FFF) (Shenzhen Jgaurora Ltd., Shenzhen, China) as shown in Figure 1c–j.

The cross section of a single filament is regarded as a circular shape with the diameter of 0.4 mm according to the FFF printer parameter requirement. In order to keep a complete 360 helicoidal cycle, all plates are printed with 24 layers that have a total thickness of 9.6 mm (the thickness of a single layer is 0.4 mm). The plate thickness is an exactly integral multiple of the filament diameter.

PLA is a thermoplastic feedstock material of the FFF printer. Being extruded to a hotbed of 60 °C along the footprint of the nozzle with a print temperature of 210 °C and a print speed of 60 mm/s, the current fused-PLA filament of each test specimen is condensed to glue the substrate—the previous condensed filaments bundle. Additionally, in the process of making test pieces, these printing parameters cannot be changed. Due to the difference of uniaxial tensile strength and glued/contact strength, the FFF printed solids are usually heterogeneous composites. Based on the form of the G-code that guides the footprint of the nozzle, we customize the filament patterns as shown in Figure 1c–j, other than the default settings of the printer.

### 2.2. Low-Velocity Impact Testing

The impact test has been carried out using the CLC-AI drop tower impact machine (Beijing Guance Co., Ltd., Beijing, China) as shown in Figure 2, which includes the automatic drop height control system, an instrumented impactor, a velocity measurement transducer, a pneumatic specimen clamp, a test fixture holding the specimens, a data collecting system, and a safety enclosure. The testing completely follows the Standard ASTM D2444. The machine performs the following sequence of operations: (1) automatically lifts drop impactor according to the set initial velocity; (2) releases magnetic force and the impactor falls; (3) impacts the specimen; (4) collects velocity, force etc. per 2 microseconds. The steel impactor weighs 2.35 kg with a semi-spherical head of 16 mm diameter. The impact velocity is set initially as 4 and 5 m/s, which means that the impactor is automatically lifted with the appropriate height. Before impact testing, the specimen is clamped on the supporter with a circular hole of 70 mm diameter. The center of the specimen completely coincides with the center of the hole. The supporter ensures complete energy conversion during the collision.

### 2.3. Numerical Simulation

In order to quantitatively investigate the impact resistance evolution has with helicoidal angle rise, we have implemented the numerical simulation by using the nonlinear finite element code, LS-Dyna [41,42]. The finite element model consists of two parts: the impactor and the PLA circular plates, which are discretized using hexahedral elements. The impactor is made up of the rigid steel with Young’s modulus *E* = 200.0 GPa, Poisson’s ratio *v* = 0.25, density *ρ* = 7850.0 kg/m^3^, yield strength *σ_ys_* = 200.0 MPa, failure strength *σ_fs_* = 200.0 MPa, and is set as the initial velocity of 4.0 or 5.0 m/s. The circular plates use the MAT ORIENTED CRACK model, which is specially used to describe the failure of brittle materials such as ceramics. In addition, by testing 3D printed standard specimens, the PLA is featured by Young’s modulus *E* = 2.0 GPa, Poisson’s ratio *v* = 0.33, density *ρ* = 1172.0 kg/m^3^, yield strength *σ_ys_* = 30 MPa, and failure strength *σ_fs_* = 30.0 MPa. To simulate the contact between the impactor and the plate, the “automatic_nodes_to_surface” contact with dynamic friction coefficients of 0.15 is used, respectively. In the ring area, as shown in Figure 1b, with an inner diameter of 7 cm on the bottom surface of the plate, the downward movement of the plate is restricted. The PLA plates’ state equation [43] is set as the Gruneisen equation of state with cubic shock velocity–particle velocity, which defines pressure for compressed materials as:(1)$$p=\frac{\rho_{0}C^{2}\mu \left[1+\left(1-\frac{\gamma_{0}}{2}\right)\mu -\frac{a}{2}\mu^{2}\right]}{{\left[1-\left(S_{1}-1\right)\mu -S_{2}\frac{\mu^{2}}{\mu +1}-S_{3}\frac{\mu^{2}}{{\left(\mu +1\right)}^{2}}\right]}^{2}}+\left(\gamma_{0}+a\mu \right)E$$
and for expanded materials as:(2)$$p=\rho_{0}C^{2}\mu +\left(\gamma_{0}+a\mu \right)E$$
where *C* is the intercept of the *v_s_*–*v_p_* curve; *S*_1_, *S*_2_, and *S*_3_ are the coefficients of the slope of the *v_s_*–*v_p_* curve;$\;\gamma_{0}$ is the Gruneisen gamma; a is the first order volume correction to $\gamma_{0}$; and $\mu =\frac{\rho }{\rho_{0}}-1$.

The plate is composed of a helical composite material, each layer is isotropic, and each layer rotates at the same angle (α) as the previous one in the vertical direction. Then Equation (3) is the constitutive equation of the first layer, and Equation (4) is the constitutive equation of the layer of rotation angle.
{*σ*} = [*D*]{*ε*}.(3)
{*σ*} = [*T*][*D*][*T*]^T^{*ε*}.(4)
where {*σ*} is the six stress components, *σ*_*x*_, *σ*_*y*_, *σ*_*z*_, *τ*_*xy*_, *τ*_*yz*_, *τ*_*zx*_; {ε} is the six strain components, *ε*_*x*_, *ε*_*y*_, *ε*_*z*_, *γ*_*xy*_, *γ*_*yz*_, *γ*_*zx*_; [*D*] is the elasticity matrix; and [*T*] is the angle transformation matrix.

## 3. Results and Discussion

### 3.1. Constituent Performance

In order to investigate the difference of the axial tensile strength and the contact strength of filaments, we fabricated three types of the standard tensile specimens (dogbone) including only-vertical-filaments (all filaments are parallel to the long side of the dogbone), only-horizontal-filaments (all filaments are parallel to the cross section of the dogbone), and default-style filaments (every single laminate consists of same-direction filaments, and up-and-down adjacent laminates are perpendicular to each other and have a 45° on the long side of dogbone.) According to the standard method, ASTM D638-039, we have carried out the static uniaxial tensile testing and the stress–strain curves are listed in Figure 3, which indicates that the PLA belongs to quasi-brittle material. That is, there is no obvious plastic yield stage before brittle damage. At the stage of strain rate smaller than 0.02, the stress/strain ratio is relatively plain. During the stage of 2.5% to 18.0% strain rate, the stress–strain curve behaves linearly to the elastic property. When the strain is close to 15.0% or more, peak tensile strengths occur among three types of dogbones. These are 34.433 MPa (grid), 60.458 MPa (contact), and 80.489 MPa (filament), respectively. It is noteworthy that static contact tensile strength of PLA is 20.031 MPa weaker than axial filament, as is Young’s modulus (YM). The YM of axial filament is 523.637 MPa, while the contact is 373.025 MPa, and the cross grid mode is 227.323 MPa.

Compared with others, there is no obvious brittle failure of the cross grid mode, which is due to the different degree of tensile failure of different wires due to the spiral angle.

### 3.2. Overall Impact Situation

In order to expose the underlying mechanism of the enhanced impact resistance, we have conducted low-velocity impact tests to monitor the responses of different helicoidal angles (see Figure 1). As shown in Figure 4, the dynamic responses include the evolution of contact force, displacement, and absorbed energy. All the specimens are damaged under an impact velocity of 4 and 5 m/s. The damaged patterns include perforation, split, and fracture (Figure 5). Although the impact resistance of PLA plates is lower than metallic or carbon fiber composites, different helicoidal circular plates have different behaviors.

From the overall destruction pattern, the curves in Figure 4 indicate that the plates are damaged brittlely. The 30°, 45°, 60° helicoidal circular plates behave better than the 15°, 75°, and 90° plates in contact force-time, contact force-displacement, and absorbed energy-time curves, which show that the helicoidal angle does affect the low-velocity impact resistance. The 30–60° helicoidal circular plates have better impact resistance than the 15°, 75° and 90° ones. However, the 0° plate is exceptional. Because the PLA plates are brittle materials, they manifest wild oscillations under the impact before being completely damaged. Therefore, contact force-time and contact force-displacement curves have peaks and troughs within 2 ms or so when they undergo asymptotic failure, as shown in Figure 4a–d. It is noteworthy that absorbed energy-time curves further indicate that 15°, 75° and 90° helicoidal plates absorb lower energy than 30°, 45°, and 60° plates, which coincide with the mechanism of bio-inspired composites. Compared with helicoidal deposition of the Stomatopod Dactyl club, our FFF-printed helicoidal composites have such inevitable shortcomings as gaps [44] existing on junction of inter-laminates and inter-filaments. Although appropriate helicoidal angles potentially enhance isotropy, they also weaken the contact area of inter-laminates or filaments, which causes discrete points of contact in the helicoidal angles and in the line contact in the 0° helicoidal plates (as shown in Appendix A). Therefore, 0° plates always have better impact resistance than other helicoidal plates, which does not conflict with the enhanced impact resistance of appropriate helicoidal composites.

Since all the specimens completely failed once, there are only one set of data curves of all specimens in Figure 5a–d. At the beginning stage, the peak contact force increases as the oscillation is ongoing. On the third oscillation, it peaks. After the third oscillation, the peak contact force gradually decays until it reaches zero. This mode occurs in all the impact testing. The dynamic process can be seen in videos SMV 1–7, as shown in the Appendix A. The maximum contact forces of the 30°, 45°, and 60° plates are obviously higher than the 15°, 75° and 90° plates, making them the best energy absorbers. Therefore, the optimal helicoidal angle does enhance low-velocity impact resistance to effectively utilize the material strength limit.

During the low-velocity impact process, the absorbed energy of all specimens monotonically increases until they are damaged. There is no rebound process unlike with carbon or metal fiber composites. The kinetic energy from the impactor gradually converts into elastic potential energy and this energy is absorbed by laminated plates, according to energy conservation. However, they are completely dissipated by various failure modes, such as filaments shear or pressure-bending fracture, cracking, and delamination. It is evident that the absorbed energy of the 30°, 45°, and 60° plates is larger than the 15°, 75° and 90° plates, which indicates that some smaller helicoidal angles are more optimal than larger ones.

Figure 5 and Figure 6 report the front and back of the specimens after collision subjected to 5 m/s velocity and numerical simulation. The failure patterns cover perforation, split and fracture. The perforated holes are characterized by a circle-like shape, and a smaller front and larger back as shown Figure 5b–e, which also shows that plates’ failure includes delamination of inter-layers and the shear or press-bending damage of in-layers. Figure 6 shows experimental and numerical failure of the 75° and 90° helicoidal plates. The experiments are in agreement with the numerical simulation where the plate stress is concentrated.

It can be clearly seen that in Figure 5a, the 0° helicoidal plate has a distinct split along the filaments laying direction. The split extends from the impact zone to the two ends of the boundary, while the filaments within the impact zone are basically fractured, and the laminates are cracked. It can be seen that in Figure 5b, the 15° helicoidal plate has an obvious pit in the impact zone, and on the back, the filaments and laminates are damaged. Additionally, almost all of the layers of the filaments are broken and lifted up, the form of damage is mainly filament breakage and delamination, and there was also damage caused by partial shear or pressure-bending. The 30°, 45°, and 60° plates in Figure 5c–e can be seen to have almost the same damage phenomena as the 15° plate. Differently, the 15° plate can witness relatively larger damage thorough the fracture hole. Figure 5f reports mainly delaminated and pressure-bending failure and smaller damage through the fracture hole in the 75° plate. It is noteworthy that, in Figure 6a, the 90° helicoidal plate shows significant cross splits along the filaments laying directions. The cross splits extend from the impact zone to the two ends of the boundary, and the intersection of the two splits lies in the center of both the plate and the impact zone. The filaments nearby the impact zone are circularly fractured in pressure-bending and are delaminated; these coincide with the numerical simulation of the isotropic circular plate with 60% effective yield strength of PLA filament and Video SMV 7-10 (see Appendix A). In Figure 6b and Video SMV 11-12 (Appendix A), there is no obvious crack on the 75° helicoidal plate; the stress is radiated around the punched hole where the contact of the impactor is, and some shredded material has flown out. Because we do not consider interfacial behavior of laminates, our numerical results do not show the delaminating process. However, the failure pattern indicates the agreement with the testing of the 90° helicoidal laminate plate, which proves that the helicoidal angle does improve the quasi-isotropy of laminates. Notably, although numerical simulation matches the failure pattern, it does not work out the oscillation.

According to the circular thin plate theory, under dynamic impact, the affected zone is divided into four different sub-zones, namely contact impactor, elastic flexural, only elastic wave, and out-of-impact. Among all the failure modes, delamination plays a main role in affecting the impact resistance of helicoidal plates. Delamination divides the plates into several separate laminates to bear the impact. More serious delamination is apt to appear shear failure such as in the 75° or 90° plate. Because the contact of the 0° plate paste between the wires better than other helicoidal plates, it has best impact resistance. However, the appropriate helicoidal plates, in the post-delamination stage, have better unity than those with larger angles. This is because the appropriate angle can likely prevent I-type fracture, which may avoid the elastic flexural zone being split into separate cantilever beam-like zones, due to the strength of the filaments slightly outnumbering the contact strength. This is the reason why the impact resistance of the 15° helicoidal plate is the best. It has an appropriate helicoidal angle which improves the quasi-isotropy of laminates. Additionally, the cantilever beam formed by the filaments enhances the low-velocity impact resistance.

## 4. Conclusions

In summary, we have demonstrated the dynamic response of helicoidal composites subjected to the low-velocity impact, and exploited the helicoidal angle between laminates to obtain potential enhanced impact resistance. Considering that fused filament fabrication (FFF) with anisotropic property is similar to bio-inspired composites, we have fabricated the specimens by a polymer FFF printer. A series of low-velocity impact tests show that adjusting the interlaminate angle of helicoidal composites definitely enhance low-velocity impact resistance. With a zero angle, the extent of failure observed in the specimens indicates viciously anisotropic. However, with a nonvanishing angle, the failure patterns of the circle-like are approximately isotropic. As the angle decreases from 60° to 30°, the impact resistance capacity becomes stronger. Through systematic validations on numerical simulations and a low-velocity impact test, we have simultaneously presented that appropriate helicoidal angles do enhance low impact resistance, and the 15° helicoidal plate is the best. Their underlying formation mechanism lies mainly in the approximate isotropy of the appropriate helicoidal angle. Our findings provide a new routine to enhance the low-velocity impact resistance for potential applications in the impact resistance design, model or test of spacecraft, aircraft, helmet, container, vehicles, tunnel, bridge, and dam in aerospace, automotive, pressure vessel, defense and civil engineering. Ongoing works include implementing the strategy to the realistic elastic-plastic or rigid-plastic carbon fiber or metallic filament PLA composites to explore more enhanced space of impact resistance.

See the Appendix A for the contact pattern of laminates and impact process videos of all specimens in the experiment and simulation.

## Figures and Tables

**Figure 1 polymers-14-01440-f001:**
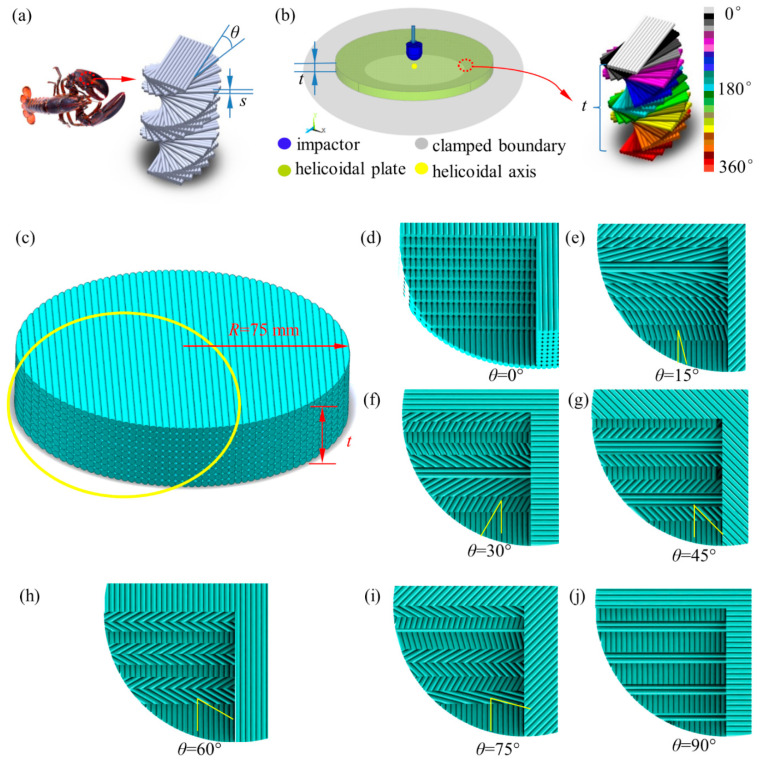
The description of helicoidal composites and their low-velocity impact. The circular plates are featured by the radius *R*, the thickness *t* of the circular plate, the thickness *s* of the single laminate, and the helicoidal angle of the interlaminates *θ*. (**a**) The Stomatopod Dactyl club and their microscale structure. (**b**) Low-velocity impact and helicoidal structure. (**c**) The outline of the specimen with the 0° helicoidal angle. Imaginary images of specimens with different helicoidal angles from (**d**) *θ* = 0°, (**e**) *θ* = 15°, (**f**) *θ* = 30°, (**g**) *θ* = 45°, (**h**) *θ* = 60°, (**i**) *θ* = 75°, and (**j**) *θ* = 90°.

**Figure 2 polymers-14-01440-f002:**
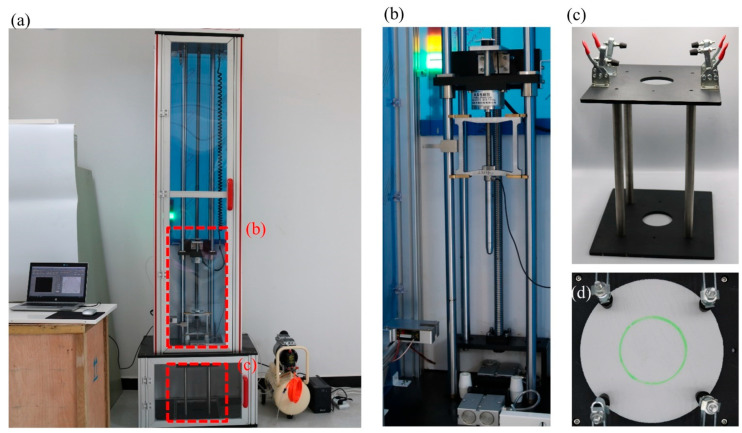
Drop tower testing machine to perform low-velocity impact testing. (**a**) Drop tower testing machine includes the automatic drop height control system, an instrumented impactor, a velocity measurement transducer, a pneumatic specimen clamp, a test fixture holding the specimens, a data collecting system, a safety enclosure, an air pressure machine and an uninterruptible power supply (UPS). (**b**) The instrumented impactor consists of a magnetic panel and a dropping head. When the magnetic force disappears, the impactor will fall as gravity accelerates to impact the specimen, which is held on a fixture. (**c**) The fixture is used to clamp circularly the specimen in (**d**).

**Figure 3 polymers-14-01440-f003:**
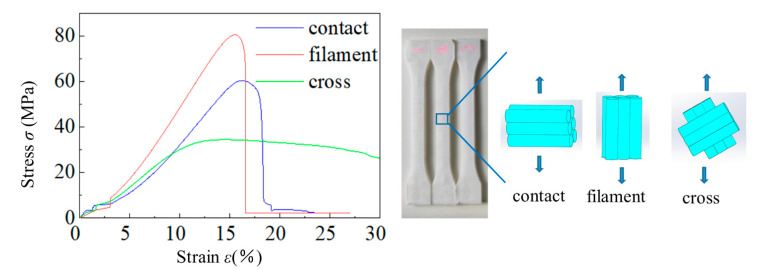
The stress–strain curves of three types of standard test pieces in the shape of dogbones, including contact, filament and cross. The contact is used to obtain the contact strength of inter-filament. The filament specimen is designed to test the uniaxial tensile strength of filament and the cross is aimed to get the strength of the mixed mode.

**Figure 4 polymers-14-01440-f004:**
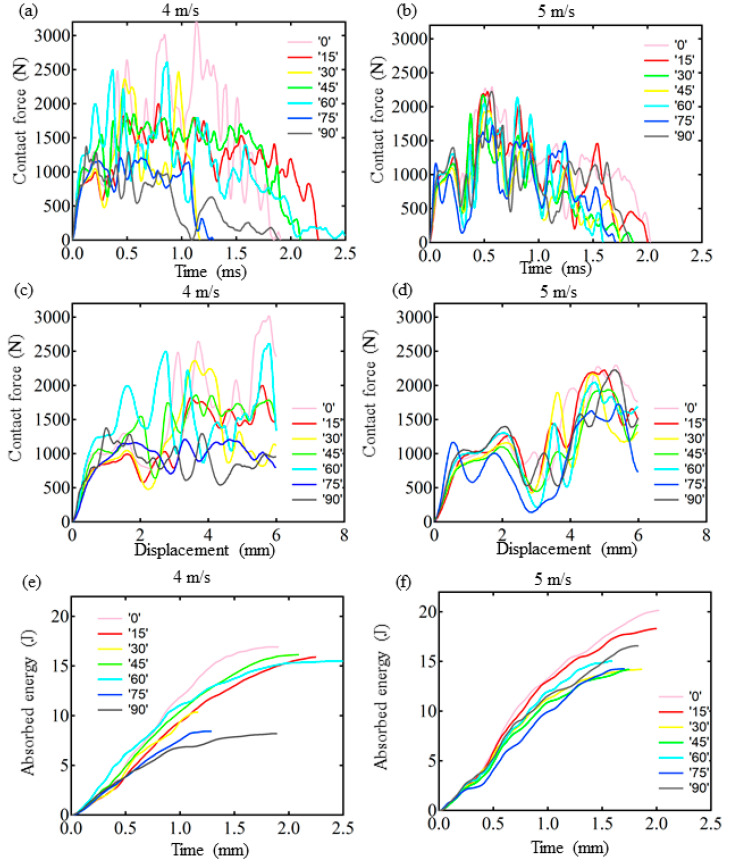
The contact force-time (**a**), contact force-displacement (**c**) and absorbed energy-time (**e**) curves of the specimens under 4 m/s impact tests. and the contact force-time (**b**), contact force-displacement (**d**) and absorbed energy-time (**f**) curves of the specimens under 5 m/s impact tests.

**Figure 5 polymers-14-01440-f005:**
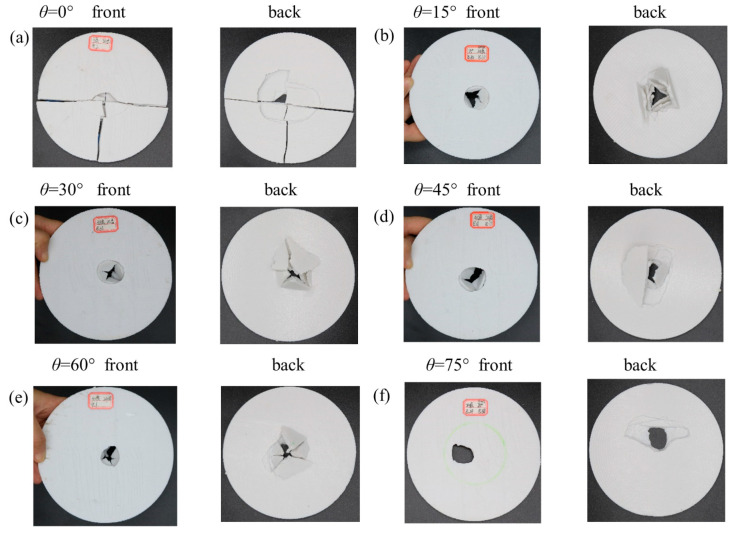
Failure modes. The front and back views of damaged patterns of 0° (**a**), 15° (**b**), 30° (**c**), 45° (**d**), 60° (**e**), and 75° (**f**) helicoidal circular plates with 24 printing layers under 5 m/s impact test.

**Figure 6 polymers-14-01440-f006:**
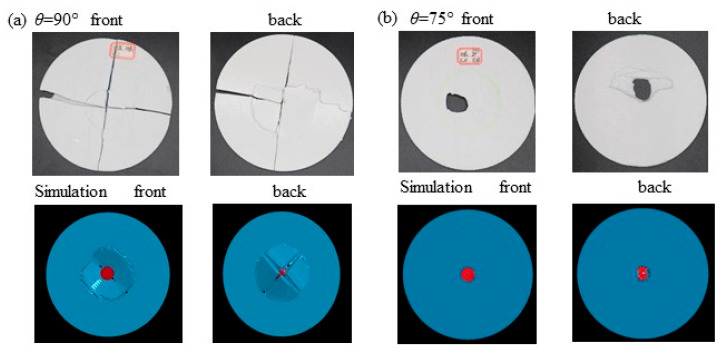
(**a**) Experimental and numerical failure of the 90° helicoidal plate. (**b**) Experimental and numerical failure of the 75° helicoidal plate.

## Data Availability

The data is all in the article.

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
