# Peer review of "Enhanced Low-Velocity Impact Resistance of Helicoidal Composites by Fused Filament Fabrication (FFF)"

_polymers, 2022, doi:10.3390/polym14071440_

Round 1
Reviewer 1 Report
This work studies the dynamic response of helicoidal composites subject under low-velocity impact utilizing experiments and simulations.
The following are recommended to improve the quality of the manuscript:
- The authors use the FDM or FFF terms for the same process. Also, officially one Journal proposes using ISO terms in additive manufacturing, i.e., the 'material extrusion 3D printing' terminology. The reviewer's suggestion is to use the FFF term. I do not like MATEX-3D printing either FDM, a trademark of Stratasys.
- Abstract. It is not so clear the added value of this research against the literature. Also, which helicoidal angle optimizes the dynamic response? This information is critical and should be reported in the 'abstract' in the reviewer's opinion.
- Keywords: 'PLA' and 'FFF' should be inserted.
- Introduction.
(i) The authors use both FDM and FFF terminology. I suggest adopting the FFF term everywhere.
(ii) In general, I like the introduction. It is compact and comprehensive. But, the material this research utilizes is the PLA. Unfortunately, in the introduction, this is not clear. Therefore, the reviewer suggests the authors include a paragraph with the FFF 3D-printing effects on PLA material, the effects of 3D printing parameters, and the optimized parameters used in this research (printing temperature, printing speed, etc.). In addition, in this paragraph, the following references should be included:
https://doi.org/10.1080/10426914.2022.2032144
https://doi.org/10.3390/jmmp4020047 - Materials and methods
Please use the FFF terminology.
Please report the 3D printing key parameters (e.g., printing speed, printing temperature, etc.) and if they are kept constant for all 3D printings. - Section 3.1. This section is compact and well structured.
- Figure 4 needs better quality.
!!!! Please note that the comments are intended merely to assist the authors in improving the paper and ensuring that published papers are of the highest quality.
Author Response
Thank you for handling the review of our paper, please see the attachment.

Reviewer 2 Report
The manuscript primarily reports a combined experimental and simulation study of low-velocity impact resistance performance of fused filament fabricated parts printed using helicoidal toolpath strategy. The PLA parts were simulated as well tested experimentally. The paper brings out and discusses key aspects of the helicoidal design in FFF and its influence on part performance.
- Page #1 (line: 41): “Columbia disaster is a typical example” - Please provide a reference.
- Page #4; Numerical Simulation:
> Please mention/ describe the material model/ MAT CARD and boundary conditions used.
> How do you take into account the anisotropy present in the printed part?
> Also, please show how the helicoidal design is modeled in the FE model.
> Generally steel has Poisson's ratio in the range of 0.27 - 0.33.
> PLA material properties: Please provide a reference. Were these properties measured on injection molded specimens or 3D printed ones?
> Please provide references for equations (1) & (2).
- Page #9 (line: 240-241): “Figure 6 shows a good agreement in experimental and numerical failure of 90 helicoidal plate”: Simulation didn't predict the crack propagation through the part experienced in the experimental testing. Please mention that for simplification, the FEA model does not take crack propagation and delamination into account, hence, such difference.
- Page #9 (line: 260-262): “But the failure pattern indicates the agreement with the testing of 90 helicoidal laminate plate, which proves that helicoidal angle does improve quasi-isotropy of laminates”: Please consider simulating an additional case where no crack was observed experimentally, e.g. 75 degree angle. Simulation vs experiment correlation in such a case can only justify this statement.
- Page #9 (line: 274-275): “This is the reason why appropriate helicoidal angles can enhance the low velocity impact resistance”: Please elaborate on which is the best angle among the ones investigated in this study, and why.
Author Response

(The authors gave the same response as above.)

Round 2
Reviewer 1 Report
The authors have improved the manuscript according to the reviewers' comments.